# Choroid Plexus Volume Change—A Candidate for a New Radiological Marker of MS Progression

**DOI:** 10.3390/diagnostics13162668

**Published:** 2023-08-14

**Authors:** Anna Jankowska, Kamil Chwojnicki, Małgorzata Grzywińska, Piotr Trzonkowski, Edyta Szurowska

**Affiliations:** 12nd Department of Radiology, Medical University of Gdańsk, Smoluchowskiego 17, 80-214 Gdańsk, Poland; eszurowska@gumed.edu.pl; 2Department of Anesthesiology and Intensive Care, Medical University of Gdańsk, Debinki 7, 80-210 Gdańsk, Poland; kchwoj@gumed.edu.pl; 3Neuroinformatics and Artificial Intelligence Lab, Department of Neurophysiology, Neuropsychology and Neuroinformatics, Medical University of Gdańsk, Debinki 7, 80-210 Gdańsk, Poland; mgrzywinska@gumed.edu.pl; 4Department of Medical Immunology, Medical University of Gdańsk, Debinki 7, 80-210 Gdańsk, Poland; ptrzon@gumed.edu.pl

**Keywords:** multiple sclerosis, choroid plexus, magnetic resonance, biomarker

## Abstract

(1) Background: Multiple sclerosis (MS) is an auto-immune, chronic, neuroinflammatory, demyelinating disease that affects mainly young patients. This progressive inflammatory process causes the chronic loss of brain tissue and results in a deterioration in quality of life. To monitor neuroinflammatory process activity and predict the further development of disease, it is necessary to find a suitable biomarker that could easily be used. In this research, we verify the usability of choroid plexus (CP) volume, a new MS biomarker, in the monitoring of the progression of multiple sclerosis disease. (2) Methods: A single-center, prospective study with three groups of patients was conducted based on the following groups: MS patients who received experimental cellular therapy (Treg), treatment-naïve MS patients and healthy controls. (3) Results: This study concludes that there is a correlation between the CPV/TIV (choroid plexus/total intracranial volume) ratio and the progress of multiple sclerosis disease—patients with MS (MS + Treg) had larger volumes of choroid plexuses. CPV/TIV ratios in MS groups were constantly and significantly growing. In the Treg group, patients with relapses had larger plexuses in comparison to the group with no relapses of MS. A similar correlation was observed for the GD+ group (patients with postcontrast enhancing plaques) compared against the non-GD group (patients without postcontrast enhancing plaques). (4) Conclusion: Choroid plexus volume, due to its immunological function, correlates with the inflammatory process in the central nervous system. We consider it to become a valuable radiological biomarker of MS activity.

## 1. Introduction

Multiple sclerosis (MS) is an auto-immune, chronic, neuroinflammatory, demyelinating disease of the central nervous system resulting in the progressive destruction of myelin sheaths, axon damage with secondary neuronal degeneration and reactive gliosis. This results in physical and cognitive disturbances in patients.

MS affects 2.5 million people worldwide with a female predominance (3:1 compared to males) [1]. Multiple sclerosis affects mainly young adults (the average onset is 30 years), frequently causing social and professional disability [2].

The cause of multiple sclerosis remains unknown. Environmental, genetic [3] and gender components as well as viral etiology were considered [4]. On the dependence of the clinical course of the disease, we can distinguish three types of MS: relapsing-remitting (RRMS), the most common, which may progress to secondary progressive disease (SPMS), and primary progressive MS, from which the minority of patients suffer [5].

The chronic and progressive inflammation of myelin and nerve fibers results in continuing brain volume loss. A large (600 patients) retrospective cohort study showed that percentage brain volume change mainly reveals the loss of white and cortical gray matter [6]. Simultaneous volumetric analysis shows that in patients with multiple sclerosis, choroid plexuses are substantially bigger than healthy controls [7].

The choroid plexuses (CPs) are located in the ventricles of the brain, form and magnify the blood–brain barrier, produce cerebrospinal fluid and serve as a port of entry for immune cells into the central nervous system [8,9,10]. In multiple sclerosis, the CP acts as an important modulator and site of inflammatory activity, as demonstrated in postmortem studies [11]. The presence of CD4 T cells, macrophages and CD11c^+^ have been confirmed [12,13,14]. Some authors claim that full knowledge about the complex role of CP in the immunological response could become a new gate for innovating therapeutic routes [8,15].

The highlight for this research is based on our belief that choroid plexus volume change can be a new radiological marker of MS progression. An additional argument here may be that CP enlargement is specific to multiple sclerosis, as it has not been observed in patients with other diseases in which an inflammatory process or demyelinating changes in the CNS are found [16].

The objective of this study was to observe if there are any differences in CP volume change in two groups of MS patients in comparison with healthy controls during the period of observation, to look for any correlations between CP volume and total brain volume, total plaques volume and the presence of active lesions.

## 2. Materials and Methods

### 2.1. Study Population

This single-center study was based on the analysis of prospectively obtained images and clinical data of 3 groups of patients:MS patients receiving cellular therapy (further marked as the Treg group);Treatment-naïve MS patients (further marked as MS group);Healthy controls (matched with MS patients according to age and sex).

The Treg group consisted of 14 patients (18–55 years) suffering from relapsing-remitting MS (RRMS) according to the revised McDonald’s criteria. There were two routes of administration of cellular drugs: intravenously (11 patients) and intrathecally (3 patients). The Treg cells were isolated from patients’ venous peripheral blood. The median of disease duration at recruitment to the study was 5 years in the intravenous group and 2 years in the intrathecal group. The patients did not receive any immunomodulating drugs for MS at least 6 months before the beginning of trial and during the follow-up.

We included in the study also two control groups—healthy controls and a treatment-naïve group. The treatment-naïve group consisted of patients (28–47 years) also with relapsing-remitting MS who had never received any immunomodulation therapy nor other MS treatment. The median of disease duration was 5 years. Healthy controls were volunteers aged 25–46 with no neurological or other impairments, without any brain lesions.

### 2.2. MRI Assessment

#### 2.2.1. Image Acquisition

MRI examinations were performed on the same scanner, a 1,5 Tesla Magnetom Aera, Siemens, Germany, with the following protocol: three-dimensional (3D) T1-weighted isovoxel precontrast sequences (slice thickness 1.50 mm, TR 1800.0 ms, TE 3.16 ms) and postcontrast sequences (Gadovist, Bayer AG, Leverkusen, Germany) and 3D fluid-attenuated inversion recovery images (FLAIR, slice thickness 1 mm, TR 6000 ms, TE 335 ms). We also performed diffusion-weighted images sequence (DWI, slice thickness 4 mm) and 3D susceptibility-weighted image sequence (SWI, slice thickness 2 mm).

MRI of the brain in the Treg group was performed at +3 months, +6 months and +12 months post-Treg administration.

Patients from the treatment-naïve group underwent brain MR on baseline and after 6 and 12 months. The healthy controls had a single examination.

#### 2.2.2. Image Postprocessing

All image assessments were conducted by two observers independently (A.J., radiologist with 13 years of experience in MR processing, and M.G., with 10 years of experience) on a Philips IntelliSpace Portal 10.

#### 2.2.3. Measurement of T2WI (T2-Weighted Images) Lesion Volume

Total number of plaques on three-dimension (3D) fluid inversion recovery sequence (Flair) was counted manually.

The plaques were manually contoured, and their volume was calculated semiautomatically.

We decided not to parcel the white matter into separate zones like Müller [17], because the majority of white matter lesions in our patients were quite advanced and confluent, and, in our opinion, it would be difficult to sort them by location.

Furthermore, we wanted to correlate the volume of plaques rather than analyze their specific locations around the ventricles. Also, we took into account the total volume of T2 lesions, not their number, as this is more reliable.

#### 2.2.4. 3DT1 Assessment

The number of postcontrast enhanced T1 lesions (Gd+) was assessed manually.

Brain structure segmentation was performed automatically on volumetric sagittal precontrast T1-weighted images by using BrainMagix software (Brussels, Belgium, all modules from 1 January 2019 to 31 August 2019).

The program counted brains’ structure volumes and their percentages of total intracranial volume (TIV).

Choroid plexuses left and right were segmented separately as presented on Figure 1, at a baseline and follow-up points.

Apart from choroid plexuses, the BrainMagix software program segmented volumes of the following structures: total brain, brain parenchyma, subcortical nuclei (caudate nuclei, thalami, globus pallidi, putamen), total white matter, cortex (with highlighted gyri in each lobe), ventricles (lateral, third, fourth), T1-hypointesities, CSF (cerebrospinal fluid) and cerebellum (with segmented cortex and white matter of each hemisphere).

### 2.3. Statistical Analysis

When comparing the volumes of choroid plexuses in the study groups, an optimal way was sought to make the result the most reliable and take into account individual patient variability. To this end, a comparison was made of the ratio: total choroid plexus volume/intracranial space volume at baseline and at the end of follow-up. The intracranial space volume was considered constant. Due to different follow-up times, the final volume was standardized to be the volume after one year of follow-up. A range of nonparametric tests was used for statistical analysis. Groups for volume differences at the same time points (independent comparisons) were compared using the U Mann–Whitney or Kruskall–Wallis tests. For dependent comparisons (assessment of volume changes over time), the sign test was used. Continuous variables were presented as means and standard deviations. *p* < 0.05 was taken as a statistically significant value.

## 3. Results

The study analyzed data from 32 people with MS and 16 healthy control subjects.

The age and gender structure of the subjects with MS and the controls is shown in Table 1. No significant differences were found between the groups.

Despite multiple segmented brain structures being processed by BrainMagix software, in our study, we focused on the correlation between CP volume and lesions which reflect MS activity: plaques volume (their changes in time) and postcontrast enhancing lesions.

As mentioned before, we decided to express choroid plexus volume as the CPV/TIV (choroid plexus volume/total intracranial volume) ratio. We agree with other authors [7,18] who emphasize that it is more reliable and allows us to minimize the individual variability of the CP volume.

This ratio measured at the start of the follow-up differed between the patients with MS and the healthy controls; the patients with MS (MS + Treg) had a larger volume of choroid plexuses (Table 2).

During 12 months of follow-up, the CPV/TIV ratios in the MS groups were constantly and significantly growing (*p* = 0.00001 in both groups, *p* = 0.0006 in MS group, *p* = 0.0005 in Treg group), as presented in Table 3.

The CPV/TIV ratio was growing similarly in both MS groups and there was no difference noticed between the sex groups (Table 4).

In the MS group, there was no difference in plexus volume between the subgroup receiving Treg and the untreated one. There was no difference in the percentage of volume change observed between the sexes.

We did not observe any significant difference in CPV/TIV between the patients receiving Tregs intravenously and intrathecally—neither at the beginning of the study (*p* = 0.65) nor after 12 months of observation (*p* = 0.22).

It was noted that in the Treg group, patients with relapses had larger plexuses compared to the group without relapses of MS. A similar observation was made regarding the GD+ group (the group of patients with postcontrast enhancing lesions) having larger plexuses compared to the GD− group (the group of patients without postcontrast enhancing lesions) (Table 4).

The annual change in the CP/TIV ratio did not correlate with either the change in brain volume or the change in plaques volume in the T2WI MRI (Figure 2 and Figure 3).

A close analysis of the Treg MS subgroup showed that there was no increase in CP volume in patients without relapses, while the CP in the Treg MS subgroup with relapses increased during the one-year follow-up (Table 5).

## 4. Discussion

In the last two decades, magnetic resonance has become an indispensable tool for the diagnosis and treatment response monitoring of patients with multiple sclerosis. The ideal MS biomarker features were described by [19] as accurate, reproducible, sensitive, correlating with and predicting relevant clinical measures and user-friendly.

The first basic MS imaging biomarkers are specific lesion profiles—they are hyperintense on T2-weighted image sequences, ovoid in shape and located in mostly periventricular and juxtacortical white matter, the corpus callosum and infratentorial areas and the spinal cord. We can also observe the postcontrast enhancement of active lesions [19]. As T2 hyperintense white matter lesions are not specific only to demyelination processes, further research was conducted to find more specific biomarkers. Among recent advances in MS neuroimaging, we can mention leptomeningeal enhancement observed on 3D FLAIR (fluid-attenuated inversion recovery) sequences, which is more suggestive of progressive rather than relapsing-remitting MS and is not observed in other disorders which can mimic MS, such as neuromyelitis optica spectrum disorders [20]. For better cortical lesion recognition, another sequence can be implemented—double inversion recovery (DIR) [21].

Furthermore, identifying chronic lesions on susceptibility-weighted images (SWI) of iron deposits indicates remyelination failure and axonal degeneration and can be a biomarker of disability and a good treatment target [22].

Also in SWI sequences, central vein signs inside white matter lesions can be detected, which is thought to be specific for MS and can differentiate MS from its mimics [23]. Unfortunately, there is a serious disadvantage to these two methods—they require an ultra-high-field MR scanner (at least 3 Tesla).

Another tool helping to assess the degree of neurodegeneration is the measurement of the brain’s structure volume change. In a large cohort study, the authors showed that there are significant differences in grey matter atrophy between phenotypes of MS, and deep gray matter volume loss drives disability progression [24].

Among experimental methods which are not broadly used in routine practice, we can list neurite orientation dispersion and density imaging (NODDI) [25], proton MR spectroscopy (1H-MRS) [26] and functional MR (fMRI) [27].

In recent years, as the immunological function of the choroid plexuses has become better known, MS researchers have started to analyze its volume correlation with disease activity and progression.

Choroid plexuses are located in the lateral, third and fourth brain ventricles. They are highly vascularized structures, composed of networks of capillaries which are fenestrated endothelia all fixed within a stroma, which is structured by cuboid epithelial cells [14]. The epithelial cells are tightly related, forming a barrier for the molecules and cells and constituting a part of the blood–brain barrier [28]. Their main and well-known function is synthesizing cerebrospinal fluid. However, choroid plexuses also have many other important functions, such as metabolic, protective, immunosurveillance and secretory (growth factors) functions.

During CNS infections and inflammatory processes, plexuses become a gateway for trafficking inflammatory cells [29]. In summary, plexuses are partially responsible for brain health.

In the last few years, multiple sclerosis scientists have emphasized the role of the choroid plexus in understanding MS pathogenesis, but other researchers also confirm its role in other neurodegenerative diseases, like the Alzheimer’s disease clinical spectrum, the brain’s aging process, Huntington’s disease and also after stroke and injuries [19,30,31,32].

An experimental mice model of disease-autoimmune encephalomyelitis (EAE) provokes immunological answers in the CP epithelium and stroma, which significantly changes the CP morphology and probably has a critical role in the communication of the immune system with the CNS [33]. Rodríguez-Lorenzo et al. proved that in the human postmortem choroid plexus stroma of progressive MS cases, -CD4+ cells, CD8+T cells and granulocytes are more numerous than in healthy controls [28]. The choroid plexus stroma also show the splitting of tight junctions in the CP epithelium [7]. In a recent study, the authors suggested that in the Alzheimer’s disease clinical spectrum, CP volume may be used as an imaging marker of mental state and cognitive function [34].

In the current study, we analyzed a group of patients suffering from relapsing-remitting MS who received in our center an experimental therapy consisting of CD4+CD25highCD127−FoxP3+ regulatory T cells. This was a clinical trial based on the observation that the activity of Treg cells is a “weak point” in the immunology systems of MS patients [35]. In our opinion, the observation of the CP volume change after the administration of Treg lymphocytes could give some answers on MS pathophysiology and also the role of CP in immunology processes.

Unfortunately, the recruited group of Treg subjects was not numerous, and for this reason, the statistical results may be—so far—underestimated. Recruitment to the study was difficult, as only patients not treated with other MS drugs were eligible for Treg administration (therefore, only 14 patients were eventually recruited).

There was no difference in CP/TIV in the groups of males and females. That is similar to the study by Ricigliano et al., where no sex dependency was also confirmed [7].

We have observed that the CPV/TIV ratio volume is significantly higher for both groups of MS patients in comparison to the reference group of healthy controls at the beginning of observation. This observation is in consistence with other authors—for example, Ricigliano et al., who reports that the CP volume ratio to the intracranial volume (CP/ICV) was 35% greater when comparing with healthy controls [7]. Also, Müller et al. observed that the CP/TIV ratio was significantly higher in MS patients than in healthy controls (21%) [17]. Interestingly, they did not report any significant difference between the CP volume in patients suffering from neuromyelitis optica spectrum disorders and the reference group. In this study, the choroid plexus volume was able to distinguish MS from NMOSD. The authors hypothesized that CP volume could be a new radiological biomarker helping to differentiate these two diseases. However, Kim et al. proved that the choroid plexus was significantly more enhanced on the postcontrast brain MRI of patients with MS as well as NMOSD [16].

In our observation time, the CPV/TIV ratio was minimally but constantly growing during one year in both groups of MS patients. This is not in agreement with other authors, who observed rather constant volumes of CP [18,36]. In our opinion, this may be due to the short time of observation in our study and also the differentiated time of disease duration in our patients. Bergsland et al. observed a stable CP volume after 5.5 years of follow-up [37]. Ricigliano et al. showed a very interesting survey where patients with clinically isolated syndromes had 29% larger choroid plexuses than healthy controls [38]. That may suggest that CPs enlarge differently according to the phase of the disease and its activity.

Furthermore, our results show that in the Treg group, the CPV/TIV ratio was essentially growing in a subgroup of patients with relapses of disease.

We focused on the clinical course of patients from the Treg group, as the treatment-naïve group was only a control group.

We separated MS Treg patients into two groups—one with relapses and one without. Interestingly, in the subgroup of patients with relapses, the CP volume was essentially growing (*p* = 0.02), but in the subgroup without relapses, the CP volume was not enlarged. Our conclusions are consistent with a study by Ricigliano et al. [7] where the authors confirmed that choroid plexus enlargement is connected to relapses in patients with MS.

We made a similar observation according to postgadolinium enhancing lesions. There was a statistically significantly higher CP volume in a group of Treg patients with at minimum one enhancing T1 lesion than in the group without active lesions. Ricigliano et al. used the ^18^F-DPA−714 uptake by white matter lesions as a marker of brain inflammation. In their study, the CP volume correlated with brain inflammation, and its value was higher for patients with gadolinium enhancing lesions [7].

Fleischer et al. [39] created the hypothesis that CP volume may mirror the activity of neuroinflammation, based on the positive correlation of choroid plexus volume with T2 lesion volume as well as disease severity and neurological disability [28]. MS researchers mostly emphasize a close association between CP volume and plaque volume. Wang et al. showed this conclusion based on a group of 99 MS patients [40]. Klistorner et al. found a strong correlation between the volume of choroid plexuses at the beginning of the study and the further development of chronic lesions [18]. Also, Müller et al. and Ricigliano et al. observed the same dependence [7,17].

This is not in line with our investigation, which showed no correlation between CPV/TIV and plaque volume. This may be due to the short observation time (other studies’ observation lasted up to 60 months) as well as relatively small groups of patients [27].

Our study had some limitations—e.g., a small number of patients due to difficulties in the recruitment process (Treg therapy remains experimental, inclusion criteria were difficult to fulfill) and the time of observation was short.

Despite these limitations, this study confirms most of the observations of other researchers. In addition, it shows a very interesting correlation—a decrease in CP as a probable effect of Treg therapy. In the future, it will probably become clear whether other MS therapies also affect CP in the same way or perhaps not.

In our study, we used commercial software for brain segmentation, but it is essential to emphasize that there are other software products already available today and that they are ready to be utilized for the automatic segmentation of choroid plexuses. This makes them very attractive tools not only for MS researchers but also in routine practice, and in consequence, the measurement of CP volume can be easily used.

## 5. Conclusions

We would like to summarize that on the basis of different MS researchers’ and neurobiologists’ findings and also our experiences, CP volume strongly reflects the inflammatory processes in multiple sclerosis and its volume change is closely correlated with disease activity (in our study with relapses of disease and the presence of active white matter lesions).

This is why, in our opinion, choroid plexus volume could become a valuable, additional radiological marker of MS progression.

## Figures and Tables

**Figure 1 diagnostics-13-02668-f001:**
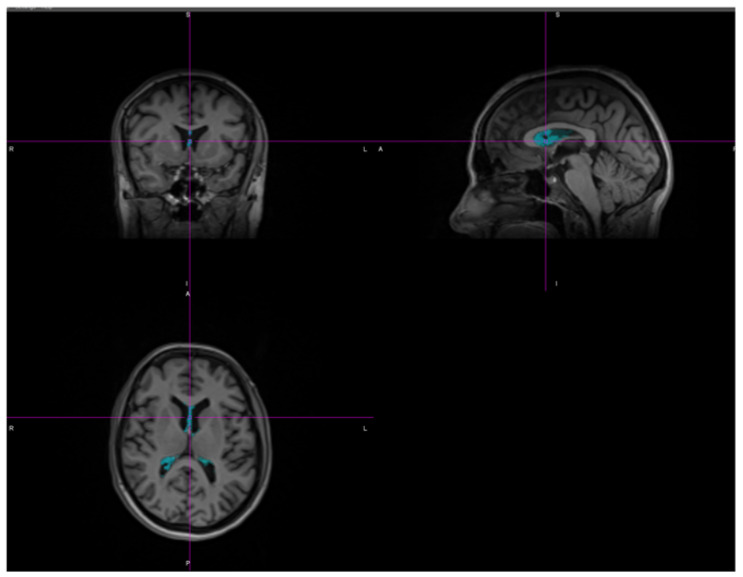
Segmented choroid plexuses in BrainMagix software.

**Figure 2 diagnostics-13-02668-f002:**
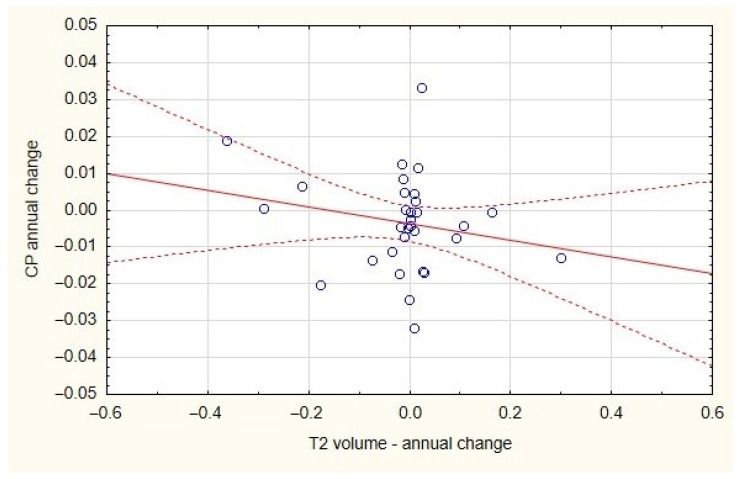
The lack of correlation between plaques volume and CP annual change for both MS groups.

**Figure 3 diagnostics-13-02668-f003:**
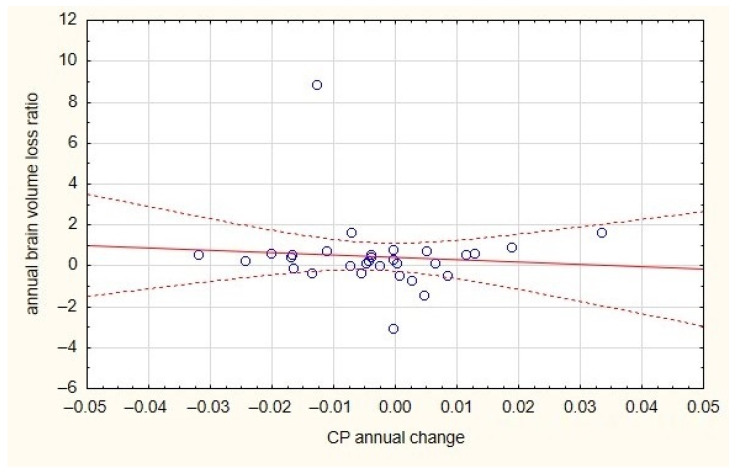
The lack of correlation between plaques volume and brain volume for both MS groups.

**Table 1 diagnostics-13-02668-t001:** Baseline characteristics of examined groups of patients.

	Group	Gender		
		Women *	Men *	All
*n*	Healthy controls	9	7	16
%		56.25	43.75	100
Age	Range 25–46	39.67 ± 6.00	36.00 ± 8.29	38.06 ± 7.09 **
*n*	MS	19	13	32
%		59.38	40.63	100
age	Range 19–51	36.26 ± 9.40	36.00 ± 7.11	36.16 ± 8.42 **
*n*	All	28	20	48

* *p* = 0.84 (chi-square test), ** *p* = 0.44 (U Mann–Whitney test).

**Table 2 diagnostics-13-02668-t002:** Baseline volume of choroid plexuses (CPs) (% of total intracranial volume—TIV).

CP—% of TIV	*n*	Mean	Std Dev	*p*
Healthy Controls	16	0.202715	0.047416	0.42 *
Treg	14	0.217707	0.027705	
MS	18	0.219945	0.050864	
All	48	0.213549	0.043891	
Controls	16	0.202715	0.047416	0.02 **
MS + Treg	32	0.218966	0.041737	

*—Kruskall–Wallis test, **—U Mann–Whitney test.

**Table 3 diagnostics-13-02668-t003:** Baseline and 12-month CP/TIV ratios expressed as % of TIV (MS + Treg, MS, Treg).

Group		*n*	Mean	Std Dev	*p* *
MS + Treg	CP—% of TIV (initial)	32	0.218966	0.041737	0.00001
MS + Treg	CP—% of TIV after 1 year of follow-up	32	0.222429	0.039983	
MS	CP—% of TIV (initial)	18	0.219945	0.050864	00006
MS	CP—% of TIV after 1 year of follow-up	18	0.223393	0.048769	
Treg	CP—% of TIV (initial)	14	0.217707	0.027705	0.0005
Treg	CP—% of TIV after 1 year of follow-up	14	0.221189	0.026437	

*—sign test.

**Table 4 diagnostics-13-02668-t004:** Annual change in CP/TIV (the difference between the initial and final volume, expressed as a percentage of TIV). GD+—the group of patients with postcontrast enhancing lesions, GD−—group of patients without postcontrast enhancing lesions.

Annual Change of CP/TIV (%)	*n*	Mean	Std Dev	*p* *
MS + Treg	32	−0.003463	0.012947	
Treg	14	−0.003482	0.010839	0.8792
MS	18	−0.003448	0.014691	
MS + Treg—women	18	−0.003428	0.015326	0.98
MS + Treg—men	14	−0.003507	0.009620	
Treg—without relapse	9	0.001413	0.009343	0.02
Treg—with ≥1 relapse	5	−0.012294	0.007501	
Treg—GD−	7	−0.005871	0.008233	0.048
Treg—with ≥1 GD+	7	−0.001094	0.013171	

*— Mann–Whitney U test.

**Table 5 diagnostics-13-02668-t005:** Change in CP volume during a one-year follow-up in relapse and no-relapse subgroups of Treg MS patients.

	*n*	CP Initial	CP 12 Month	Mean Difference (CP Initial–CP 12 Month)
Treg relapse	5 *	0.20548	0.217774	−0.012294
Treg no relapse	9 **	0.2245	0.223087	0.001413

*—*p* = 0.0436 (sign test)—significant difference between CP initial and CP 12 month (increase of CP volume) in Treg MS group with relapses; **—*p* = 0.99 (sign test)—no significant difference between CP initial and CP 12 month (volume of initial CP is equal to CP 12 month).

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
