# Peer review of "Choroid Plexus Volume Change—A Candidate for a New Radiological Marker of MS Progression"

_diagnostics, 2023, doi:10.3390/diagnostics13162668_

Round 1

Reviewer 1 Report

In multiple sclerosis (MS), the Choroid plexus (CP) plays a significant role in regulating neuroinflammation. Therefore, the study of CP can provide significant insight into this pathology. “Choroid plexus volume change - a candidate for a new radio- 2 logical markers of MS progression” by Jankowska et al discussed the correlation between the CPV/TIV ratio and the progression of multiple sclerosis disease. Importantly, they showed that patients treated with experimental cellular therapy (TREG) with relapses had larger plexuses with no relapses in MS patients. This manuscript highlights that a change in the volume of CP can be a new radiological marker of MS progression.

The manuscript is well-written with elaborate protocols. The manuscript is interesting and contributes to the field.

Minor comments:

1.       This manuscript has mentioned a number of abbreviations. Most of them are not defined. This makes it very difficult for the readers and left for readers to assume. Make sure to describe abbreviations in the abstract and introduction or results, wherever they are mentioned for the first time.  CP, TIV, GD+.

 2.       What is the ‘SM patient group’? This word is mentioned in many places in the text and figure legends but without any explanation.  

 3.       Was there any difference in CPV/ TIV ratio in between intravenously and intrathecally treated patients?

Minor editing is required in regard to punctuations used in the sentences. 

Author Response

  1. I have expanded the abreviations in the text, as you requested
  2. I have corrected the abreviation SM group to MS group
  3. There was no significant difference in CPV/TIV between intravenously and intrathecally treated patients neither  at the begining of the survey nor after one year of observation.

Reviewer 2 Report

This is interesting article descibing choroid plexus volume change as a candidate for a radiological marker of MS progression.

I have some comments mainly according to paper organization, please explain all abbreviations first time used in the text and abstract, as well as in Tables (footnotes - tables description).

In discussion, or shorty in introduction I suggest to discuss and mention other proposed radiological biomarkers of MS as well as thier significance.

Author Response

  1. I have expanded all the abreviations in the text
  2. In discusion I have mentioned other radiological biomarkers of MS as well as their significance

Round 2

Reviewer 2 Report

The authors corrected the article according to suggestions, I recommend to accept this article